# Online Self-Calibration of 3D Measurement Sensors Using a Voxel-Based Network

**DOI:** 10.3390/s22176447

**Published:** 2022-08-26

**Authors:** Jingyu Song, Joonwoong Lee

**Affiliations:** Department of Industrial Engineering, Chonnam National University, 77 Yongbong-ro, Buk-gu, Gwangju 61186, Korea

**Keywords:** online self-calibration, convolutional neural network, voxel information

## Abstract

Multi-sensor fusion is important in the field of autonomous driving. A basic prerequisite for multi-sensor fusion is calibration between sensors. Such calibrations must be accurate and need to be performed online. Traditional calibration methods have strict rules. In contrast, the latest online calibration methods based on convolutional neural networks (CNNs) have gone beyond the limits of the conventional methods. We propose a novel algorithm for online self-calibration between sensors using voxels and three-dimensional (3D) convolution kernels. The proposed approach has the following features: (1) it is intended for calibration between sensors that measure 3D space; (2) the proposed network is capable of end-to-end learning; (3) the input 3D point cloud is converted to voxel information; (4) it uses five networks that process voxel information, and it improves calibration accuracy through iterative refinement of the output of the five networks and temporal filtering. We use the KITTI and Oxford datasets to evaluate the calibration performance of the proposed method. The proposed method achieves a rotation error of less than 0.1° and a translation error of less than 1 cm on both the KITTI and Oxford datasets.

## 1. Introduction

Multi-sensor fusion is performed in many fields, such as autonomous driving and robotics. A single sensor does not guarantee reliable recognition in complex and varied scenarios [1]. Therefore, it is difficult to cope with various autonomous driving situations using only one sensor. Conversely, fusing two or more sensors supports reliable environmental perception around the vehicle. In multi-sensor fusion, one sensor compensates for the shortcomings of the other sensor [2]. In addition, multi-sensor fusion expands the detection range and improves the measurement density compared with using a single sensor [3]. Studies based on multi-sensor fusion include 3D object detection, road detection, semantic segmentation, object tracking, visual odometry, and mapping [4,5,6,7,8,9]. Moreover, most large datasets that are built for autonomous driving research [10,11,12,13] provide data measured by at least two different sensors. Importantly, multi-sensor fusion is greatly affected by the calibration accuracy of the sensors used. While the vehicle is driving, the pose or position of the sensors mounted on the vehicle may change for various reasons. Therefore, for multi-sensor fusion, it is essential to perform the online calibration of sensors to accurately recognize changes in sensor pose or changes in the positions of the sensors.

Extensive academic research on multi-sensor calibration has been performed [2,3]. Many traditional calibration methods [14,15,16,17] use artificial markers, including checkerboards, as calibration targets. The target-based calibration algorithms are not suitable for autonomous driving because they involve processes that require manual intervention. Some of the calibration methods currently used focus on fully automatic and targetless online self-calibration [3,18,19,20,21,22,23,24]. However, most online calibration methods perform calibration only when certain conditions are met, and their calibration accuracy is not as high as the target-based offline methods [1]. The latest online calibration methods [1,2,25,26,27] based on deep learning use optimization through gradient descent, large-scale datasets, and CNNs to overcome the limitations of the previous online methods. In particular, the latest research based on CNNs has shown suitable results. Compared with previous methods, CNN-based online self-calibration methods do not require strict conditions and provide excellent calibration accuracy when they are run online.

Many CNN-based LiDAR-camera calibration methods use an image for calibration. In this case, the point cloud of the LiDAR is projected onto the image. Then, 2D convolution kernels are used to extract the features of the inputs.

In this study, we propose a CNN-based multi-sensor online self-calibration method. This method estimates the values of six parameters that describe rotation and translation between sensors that are capable of measuring 3D space. The combinations of sensors that are subject to calibration in our proposed method are: a LiDAR and stereo camera and a LiDAR and LiDAR. One of the two sensors is set as the reference sensor and the other as the target sensor. In the combination of LiDAR and stereo camera, the stereo camera is set as the reference sensor.

The CNN we propose is a network that uses voxels instead of using image features. Therefore, we convert the stereo image into 3D points called pseudo-LiDAR points to feed the stereo image into this network. Pseudo-LiDAR points and actual LiDAR points are expressed in voxel spaces through voxelization. Then, 3D convolution kernels are applied to the voxels to generate features that can be used for calibration parameter regression. In particular, the attention mechanism [28] included in our proposed network confirms the correlation between the input information of the two sensors. The research fields that use voxels are diverse, including shape completion, semantic segmentation, multi-view stereoscopic vision, object detection, etc. [29,30,31,32].

The amount of data in public datasets is insufficient to perform online self-calibration. Therefore, existing studies have assigned random deviations to the values of known parameters and have evaluated the performance of online self-calibration based on how accurately the algorithm proposed in the respective study predicts this deviation. This approach is commonly referred to as miscalibration [1,2,25]. To sample the random deviation, we choose the rotation range and translation range as ±20° and ±1.5 m, respectively, as in [1]. In this study, we train five networks on a wide range of miscalibrations and apply iterative refinement to the outputs of the five networks and temporal filtering over time to increase the calibration accuracy. The KITTI dataset [10] and Oxford dataset [12] are used to conduct the research of the proposed method. The KITTI dataset is used for online LiDAR-stereo camera calibration, and the Oxford dataset is used for online LiDAR-LiDAR calibration.

The rest of this paper is organized as follows. Section 2 provides an overview of existing calibration studies. Section 3 describes the proposed method. Section 4 presents the experimental results for the proposed method, and Section 5 draws conclusions.

## 2. Related Work

This section provides a brief overview of traditional calibration methods. In addition, we introduce how CNN-based calibration methods have been improved. Specifically, the calibrations covered in this section are LiDAR-camera calibration and LiDAR-LiDAR calibration.

### 2.1. Traditional Methods

Traditional methods of calibration that use targets mainly use artificial markers. One example is the LiDAR-camera calibration method that uses a polygonal planar board [14]. This method first finds the vertices of the planar board in the image and in the LiDAR point cloud. Then, the corresponding points between the vertices of the image and the vertices of the point cloud are searched, and a linear equation is formulated. Finally, this method uses singular value decomposition to solve the linear equation and obtain calibration parameters. Another example is the LiDAR-camera calibration method that uses a planar chessboard [15]. The edge information in an image and the Perspective-n-Point algorithm are used to find the plane of the chessboard that appears in the image. Then, distance filtering and a random sample consensus (RANSAC) algorithm are used to obtain the chessboard plane information from the LiDAR point cloud. After obtaining the plane information, this method aligns the normal vectors of the planes with obtaining the rotation parameters between LiDAR and the camera. The translation parameters between LiDAR and camera are calculated by minimizing the distance between the plane searched from the LiDAR points, and the rotated plane searched from the image. Similarly, the algorithm in [16] is an automatic LiDAR-LiDAR calibration that uses planar surfaces. This method uses three planes. The planes are obtained through a RANSAC algorithm. The calibration of LiDAR-LiDAR is formulated as a nonlinear optimization problem by minimizing the distance between corresponding planes. This approach adopts the Levenberg-Marquardt algorithm for nonlinear optimization. Another LiDAR-LiDAR calibration in [17] uses two poles plastered with retro-reflective tape to easily identify them in the point cloud. This method first uses a threshold to find the reflected points on the pole. The searched point cloud is expressed as a linear equation that represents a line. The points and the linear equation are then used to solve the least squares problem. The data obtained by solving the least squares problem are the calibration parameters.

Among the targetless online LiDAR-camera calibration methods, some methods use edge information [18,19]. In these methods, when the LiDAR-camera calibration is correct, the edges of the depth map of the LiDAR points are projected onto the image, and the edges of the image are naturally aligned. As another example, there is a method that uses the correlation between sensor data, as in [20,21]. The method in [20] uses the reflectance information from the LiDAR and the intensity information from the camera for calibration. According to this method, the correlation between reflectivity and intensity is maximized when the LiDAR-camera calibration is accurate. The method in [21] uses the surface normal of the LiDAR points and the intensity of image pixels for calibration. There is also a method based on the hand-eye calibration framework [22], which estimates the motion of the LiDAR and camera, respectively, and uses this information for calibration. A targetless online LiDAR-LiDAR calibration is introduced in [23]. This method first performs a rough calibration from an arbitrary initial pose. Then, the calibration parameters are corrected through an iterative closest point algorithm and are further optimized using an octree-based method. Other LiDAR-LiDAR calibration methods are presented in [3,24]. These methods are based on the hand-eye calibration framework, and the motion of each LiDAR is estimated. This information is used for calibration.

### 2.2. CNN-Based Methods

RegNet [25], the first CNN-based online LiDAR-camera self-calibration method, adopted a three-step convolution consisting of feature extraction, feature matching, and global regression. RegNet uses the decalibration of a given point cloud to train the proposed CNN and also uses five networks to predict the six-degree of freedom (6-DoF) extrinsic parameters for five different decalibration ranges.

CalibNet [5], the CNN-based online LiDAR-camera self-calibration method, proposed a geometrically supervised deep network that was capable of automatically predicting the 6-DoF extrinsic parameters. The end-to-end training is performed by maximizing the photometric and geometric consistencies. Here, photometric consistency is obtained between two depth maps constructed by projecting a given point cloud onto the input image with the 6-DoF parameters predicted by the network and the ground-truth 6-DoF parameters. Similarly, geometric consistency is calculated between two kinds of 3D points obtained by transforming the point cloud into 3D space with the predicted 6-DoF parameters and the ground-truth 6-DoF parameters.

LCCNet [1], which represents a significant improvement over previous CNN-based methods, is a CNN-based online LiDAR-camera self-calibration method. This network considers the correlation between the RGB image features and the depth image projected from point clouds. Additional CNN-based online LiDAR-camera self-calibration methods are presented in [26,27]. They utilize the semantic information extracted by CNN to perform more robust calibration even under changes in lighting and noise [26].

To the best of our knowledge, no deep learning-based LiDAR-LiDAR or LiDAR-stereo camera online self-calibration method has yet been reported. In this paper, we propose, for the first time, a deep learning-based method that is capable of online self-calibration between such sensor combinations.

## 3. Methodology

This section describes the preprocessing of stereo images and LiDAR point clouds, the structure of our proposed network, the loss function for network training, and the postprocessing of the network output. These descriptions are commonly applied to the calibration of the LiDAR-stereo camera and LiDAR-LiDAR combinations. We chose a LiDAR as the target sensor in the LiDAR-stereo camera and LiDAR-LiDAR combinations, and the rest of the sensors as the reference sensor. 

### 3.1. Preprocessing

In order to perform online self-calibration with the network we designed, several processes, including data preparation, were performed in advance. This section describes these processes. We assume that sensors targeted for online calibration are capable of 3D measurement. Therefore, we use point clouds that are generated by these sensors. In the LiDAR-LiDAR combination, this premise is satisfied, but in the case of the LiDAR-stereo camera combination, this premise is not satisfied, so we obtain a 3D point cloud from the stereo images. The conversion of the stereo depth map to 3D points and the removal of the 3D points, which are covered in the next two subsections, are not required for the LiDAR-LiDAR combination.

#### 3.1.1. Conversion of Stereo Depth Map to 3D Points 

A depth map is built from stereo images through stereo matching. In this paper, we obtain the depth map using the method in [33] that implements semi-global matching [34]. This depth map composed of disparities is converted to 3D points, which are called pseudo-LiDAR points, as follows: (1)$$P=\left[\begin{array}{ccc} X & Y & Z \end{array}\right]=\left[\begin{array}{ccc} \frac{base\cdot \left(u-c_{u}\right)}{disp} & \frac{base\cdot \left(v-c_{v}\right)}{disp} & \frac{f_{u}\cdot base}{disp} \end{array}\right]$$
where $c_{u}$, $c_{v}$, and $f_{u}$ are the camera intrinsic parameters, u and v are pixel coordinates, base is the baseline distance between cameras, and disp is the disparity obtained from the stereo matching.

#### 3.1.2. Removal of Pseudo-LiDAR Points

The pseudo-LiDAR points are too many in number compared with the points measured by a LiDAR. Therefore, we, in this paper, reduce the quantity of pseudo-LiDAR points through a spherical projection, which is implemented using the method presented in [35] as follows:(2)$$\left[\begin{array}{c} p\\ q \end{array}\right]=\left[\begin{array}{c} \frac{1}{2}\left(1-\mathrm{atan}\left(-X,Z\right)/\pi \right)\cdot w\\ \left(1-\left(\mathrm{asin}\left(-Y\cdot r^{-1}\right)+f_{up}\right)\cdot f^{-1}\right)\cdot h \end{array}\right]$$
where (*X*, *Y*, *Z*) are 3D coordinates of a pseudo-LiDAR point, (p, q) are the angular coordinates, (h, w) are the height and width of the desired projected 2D map, r is the range of each point, and $f=f_{up}+f_{down}$ is the vertical field of view (FOV) of the sensor. We set $f_{up}$ to 3° and $f_{down}$ to −25°. Here, the range 3° to −25° is the vertical FOV of the LiDAR used to build the KITTI benchmarks [10]. The pseudo-LiDAR points become a 2D image via this spherical projection. Multiple pseudo-LiDAR points can be projected onto a single pixel in the 2D map. In this case, only the last projected pseudo-LiDAR point is left, and the previously projected pseudo-LiDAR points are removed.

#### 3.1.3. Setting of Region of Interest

Because the FOVs of the sensors used are usually different, we determine the region of interest (ROI) of each sensor and perform calibration only with data belonging to this ROI. However, the ROI cannot be determined theoretically but can only be determined experimentally. We determine the ROI of the sensors by looking at the distribution of data acquired with the sensors.

We provide an example of setting the ROI using data provided by the KITTI [10] and Oxford [12] datasets. For the KITTI dataset, which was built using a stereo camera and LiDAR, the ROI of the stereo camera is set to [Horizon: −10 m–10 m, Vertical: −2 m–1 m, Depth: 0 m–50 m], and the ROI of the LiDAR is set to the same values as the ROI of the stereo camera. For the Oxford dataset, which was built using two LiDARs, the ROI of the LiDAR is set to [Horizon: −30–30 m, Vertical: −2–1 m, Depth: −30–30 m].

#### 3.1.4. Transformation of Point Cloud of Target Sensor

In this paper, the miscalibration method used in previous studies [1,2,25] is used to perform the calibration of the stereo camera-LiDAR and LiDAR-LiDAR combination. In the KITTI [10] and Oxford [12] datasets we use, the values of six extrinsic parameters between two heterogeneous sensors and the 3D point clouds generated by them are given. Therefore, we can transform the 3D point cloud created by one sensor into a new 3D point cloud using the values of these six parameters. If we assign arbitrary deviations to these parameters, we can retransform the transformed point cloud in another space. At this time, if a calibration algorithm accurately finds the deviations that we randomly assign, we can move the retransformed point cloud to the position before the retransformation.

In order to apply the aforementioned approach to our proposed online self-calibration method, a 3D point P=x, y,z∈ℝ3 measured by the target sensor is transformed by Equation (3) as follows: (3)P′^=RTmisRTinitPT^=Rmis000Tmis1Rinit000Tinit1PT^
(4)RTgt=RTmis−1
(5)Rinit=cosRz−sinRz0sinRzcosRz0001cosRy0sinRy010−sinRy0cosRy1000cosRx−sinRx0sinRxcosRx
(6)Tinit=TxTyTzT
(7)Rmis=cosθz−sinθz0sinθzcosθz0001cosθy0sinθy010−sinθy0cosθy1000cosθx−sinθx0sinθxcosθx
(8)Tmis=τxτyτzT
where *P*′ is the transformed point of *P*, and superscript T represents the transpose. P^ and P′^ are expressed with homogeneous coordinates. RTgt, described in Equation (4), is the transformation matrix we want to predict with our proposed method. RTgt is used as the ground truth when the loss for training is calculated. In Equation (5), each of the parameters Rx, Ry, and Rz describes the angle rotated about the *x*-, *y*-, and *z*-axes between the two sensors. In Equation (6), Tx, Ty, and Tz describe the relative displacement between two sensors along the *x*-, *y*-, and *z*-axes. In this study, we assume that the values of the six parameters Rx, Ry, Rz, Tx, Ty, and Tz are given. In Equations (7) and (8), the parameters θx, θy, θz, τx, τy, and τz represent the random deviations for Rx, Ry, Rz, Tx, Ty, and Tz, respectively. Each of these six deviations is sampled randomly with equal probability within a predefined range of deviations described next. In Equations (5)–(8), Rinit and Rmis are rotation matrices and Tinit and Tmis are translation vectors. The transformation by Equation (3) is performed only on points belonging to a predetermined ROI of the target sensor.

We set the random sampling ranges for $\theta_{x}$, $\theta_{y}$, $\theta_{z}$, $\tau_{x}$, $\tau_{y}$, and $\tau_{z}$ the same as in previous studies [1,25] as follows: (rotational deviation: −θ–θ, translation deviation: −τ–τ), Rg1 = {θ: ±20°, τ: ±1.5 m}, Rg2 = {θ: ±10°, τ: ±1.0 m}, Rg3 = {θ: ±5°, τ: ±0.5 m}, Rg4 = {θ: ±2°, τ: ±0.2 m}, and Rg5 = {θ: ±1°,τ: ±0.1 m}. Each of Rg1, Rg2, Rg3, Rg4, and Rg5 set in this way is used for training each of the five networks named Net1, Net2, Net3, Net4, and Net5. One deviation range is assigned to one network training. Training for calibration starts with Net1 assigned to Rg1, and it continues with networks assigned to progressively smaller deviation ranges. The network mentioned here is described in Section 3.2.

#### 3.1.5. Voxelization

We first perform a voxel partition by dividing the 3D points obtained by the sensors into equally spaced 3D voxels, as was performed in [36]. This voxel partition requires a space that limits the 3D points acquired by a sensor to a certain range. We call this range a voxel space. We consider the length of a side of a voxel, which is a cube, as a hyper-parameter, and denote it as S. In this paper, the unit of S is expressed in cm. A voxel can contain multiple points, of which up to three are randomly chosen, and the rest are discarded. Here, it is an experimental decision that we leave only up to three points per voxel. Referring to the method in [37], the average coordinates along the x-, y-, and z-axes of the points in each voxel are then calculated. We build three initial voxel maps, *F_x_*, *F_y_*, and *F_z_*, using the average coordinates for each axis. For each sensor, these initial voxel maps become the input to our proposed network. Section 3.2 describes the network. 

In this paper, we set the voxel space to be somewhat larger than the predetermined ROI of the sensor, considering the range of deviation. For example, in the case of the KITTI dataset, the voxel space of the stereo input is set as [horizontal: −15–15 m, vertical: −15–15 m, depth: 0–55 m], and the voxel space of the LiDAR input is set to the same size as the voxel space of the stereo input. In contrast, the voxel space of the 3D points generated by the two LiDARs in the Oxford dataset is set to [width: −40–40 m, height: −15–15 m, depth: −40–40 m]. The points outside of the voxel space are discarded. 

### 3.2. Network Architecture

We propose a network of three parts, which are referred to as a feature extraction network (FEN), an attention module (AM), and an inference network (IN). The overall structure of the proposed network is shown in Figure 1. The input of this network is the *F_x_*, *F_y_*, and *F_z_* for each sensor built from voxelization, and the output is seven numbers, three of which are translation-related parameter values, and the other four are rotation-related quaternion values. The network is capable of end-to-end training because every step is differentiable.

#### 3.2.1. FEN

Starting from the initial input voxel maps *F_x_*, *F_y_*, and *F_z_*, FEN extracts features for use in predicting calibration parameters by performing 3D convolution on 20 layers. The number of layers, the size of the kernel used, the number of kernels used in each layer, and the stride applied in each layer are experimentally determined. The kernel size is 3 × 3 × 3. There are two types of stride, 1 and 2, which are used selectively for each layer. The number of kernels used in each layer is indicated at the bottom of Figure 1. This number corresponds to the quantity of the feature volume created in the layer. In the deep learning terminologies, this quantity is called channels. Convolution is performed differently depending on the stride applied to each layer. When stride 1 is applied, submanifold convolution [38] is performed, and when stride 2 is applied, general convolution is performed. General convolution is performed on all voxels with or without a value, but submanifold convolution is performed only when a voxel with a value corresponds to the central cell of the kernel. In addition, batch normalization (BN) [39] and rectified linear unit (ReLU) activation functions are sequentially applied after convolution in the FEN.

We want the proposed network to perform robust calibration for large rotational and translational deviations between two sensors. To this end, a large receptive field is required. Therefore, we included seven layers with a stride of 2 in the FEN. 

The final output of the FEN is 1024 feature volumes. The number of cells in the feature volume depends on the size of the voxel, but we let V be the number of cells in the feature volume. At this time, because each feature volume can be reconstructed as a V-dimensional column vector, we represent 1024 feature volumes as a matrix F of dimension V × 1024. The outputs of FENs for the reference and target sensors are denoted by *F_r_* and *F_t_*, respectively.

#### 3.2.2. AM

It is not easy to match the features extracted from the FEN through convolutions because the point clouds from the LiDAR-stereo camera combination are generated differently. Even in the LiDAR-LiDAR combination, if the FOVs of the two LiDARs are significantly different, it is also not easy to match the features extracted from the FEN through convolutions. Moreover, because the deviation range of rotation and translation is set large to estimate calibration parameters, it becomes difficult to check the similarity between the point cloud of the target sensor and the point cloud of the reference sensor.

Inspired by the attention mechanism proposed by Vaswani et al. [28], we solve these problems: we design an AM that implements the attention mechanism, as shown in Figure 1. The AM calculates an attention value for each voxel of the reference sensor input using the following procedure. 

The AM has four fully connected layers (FCs): FC_1_, FC_2_, FC_3_, and FC_4_. A feature is input into these FCs, and a transformed feature is output. We denote the outputs of FC_1_, FC_2_, FC_3_, and FC_4_ as matrices M_1_, M_2_, M_3_, and M_4_, respectively. Each FC has 1024 input nodes. Here, the number 1024 is the number of feature volumes extracted from the FEN. The FC_1_ and FC_4_ have G/2 output nodes, and the FC_2_ and FC_3_ have G output nodes. These FCs transform 1024 features to G or G/2 features. Here, G is a hyper-parameter. If the sum of the elements in a row of matrix F, which is the output of the FEN, is 0, the row vector is not input to FC. We apply layer normalization (LN) [40] and the ReLU function to the output of these FCs so that the final output becomes nonlinear. The output M_2_ of FC_2_ is a matrix of dimension *V_t_* × G, and the output M_3_ of FC_3_ is a matrix of dimension *V_r_* × G. Here, *V_r_* and *V_t_* are the number of rows in which there is at least one element with a feature value among the elements in each row of *F_r_* and *F_t_*, respectively. Therefore, *V_r_* and *V_t_* can be different for each input. However, we fix the values of *V_r_* and *V_t_* because the number of input nodes of the multi-layer perceptron (MLP) of the IN following the AM cannot be changed every time. In order to fix the values of *V_r_* and *V_t_*, we input all the data to be used in the experiments into the network and set the values when they are the largest, but we make them a multiple of 8. This is because *V_r_* and *V_t_* are also hyper-parameters. If the actual *V_r_* and *V_t_* are less than the predetermined *V_r_* and *V_t_*, the elements of the output matrices of FCs will be filled with zeros. The output M_1_ of FC_1_ is a matrix of dimension *V_t_* × G/2, and the output M_4_ of FC_4_ is a matrix of dimension *V_r_* × G/2.Computation of attention score by dot product

An attention score is obtained from the dot product of a row vector of M_3_ and a column vector of $M_{2}^{T}$. This score is the same as the cosine similarity. The matrix A_S_ is obtained through the dot products of all row vectors of M_3_ and all column vectors of $M_{2}^{T}$ are called an attention score matrix. The dimension of the matrix A_S_ (A_S_ = M_3_$\cdot M_{2}^{T}$) is *V_r_* × *V_t_*.Generation of attention distribution by softmax

We apply the softmax function to each row vector of A_S_ and obtain the attention distribution. The softmax function calculates the probability of each element of the input vector. We call this probability an attention weight, and the matrix obtained by this process is the attention weight matrix A_W_ of dimension *V_r_* × *V_t_*.Computation of attention value by dot product

An attention value is obtained from the dot product of a row vector of A_W_ and a column vector of the matrix M_1_. A matrix A_V_ obtained through the dot products of all row vectors of A_W_ and all column vectors of M_1_ is called an attention value matrix. The dimension of the matrix A_V_ (A_V_ = A_W_·M_1_) is *V_r_* × G/2.

Finally, we concatenate the attention value matrix A_V_ and the matrix M_4_. The resulting matrix from this final process is denoted as A_C_ (A_C_ = [A_V_ M_4_]) and has dimension *V_r_* × G; this matrix becomes the input to the IN. The reason we set the output dimension of FC_1_ and FC_4_ to G/2 instead of G is to save memory and reduce processing time.

#### 3.2.3. IN

The IN infers rotation and translation parameters. The IN consists of an MLP and two fully connected layers, FC_5_ and FC_6_. The MLP is composed of an input and an output layer, as well as a single hidden layer. The input layer has *V_r_* × G nodes, and the hidden and output layers have 1024 nodes, respectively. Therefore, when we input A_C_, the output of the AM, into the MLP, we make A_C_ a flat vector. In addition, this MLP has no bias input, and it uses ReLU as an activation function. Moreover, LN is performed on the weighted sums that are input to nodes in the hidden layer and output layer, and ReLU is applied to the normalization result to obtain the output of these nodes. The output of the MLP becomes the input to the FC_5_ and FC_6_. The MLP plays the role of dimension reduction in the input vector.

We do not apply a normalization or an activation function to the FC_5_ and FC_6_. FC_5_ produces three translation-related parameter values, which are $\tau_{x}^{p}$, $\tau_{y}^{p}$, and $\tau_{z}^{p}$, and FC_6_ produces four rotation-related quaternion values, which are *q*_0_, *q*_1_, *q*_3_, and *q*_4_.

### 3.3. Loss Function

To train the proposed network, we use a loss function as follows: (9)$$L=\lambda_{1}L_{rot}+\lambda_{2}L_{trs}$$
where $L_{rot}$ is a regression loss related to rotation, $L_{trs}$ is a regression loss related to translation, and hyper-parameters *λ*_1_ and *λ*_2_, respectively, are their weights. We use the quaternion distance to regress the rotation. The quaternion distance is defined as: (10)$$L_{rot}=\mathrm{acos}(2{\left(\frac{q_{p}}{\left|q_{p}\right|}\cdot \frac{q_{gt}}{\left|q_{gt}\right|}\right)}^{2}-1)$$
where · represents the dot product, |·| indicates the norm, and $q_{p}$ and $q_{gt}$ indicate a vector of the quaternion parameters predicted by the network and the ground-truth vector of quaternion parameters, respectively. From $RT_{gt}$ of Equation (4), we obtain the four quaternion values. These four quaternion values are used for rotation regression as the ground truth. 

For the regression of the translation vector, the smooth *L*1 loss is applied. The loss $L_{trs}$ is defined as follows: (11)$$L_{trs}=\frac{1}{3}\left(smooth_{L1}\left(\tau_{x}^{p}-\tau_{x}^{gt}\right)+smooth_{L1}\left(\tau_{y}^{p}-\tau_{y}^{gt}\right)+smooth_{L1}\left(\tau_{z}^{p}-\tau_{z}^{gt}\right)\right)\;smooth_{L1}\left(x\right)=\left\{\begin{array}{c} \begin{array}{cc} \frac{x^{2}}{2\beta } & \;\mathrm{if}\;\left|x\right|<\beta \end{array}\\ \begin{array}{cc} \left|x\right|-\frac{\beta }{2} & \;\mathrm{otherwise} \end{array} \end{array}\right.$$
where the superscripts *p* and *gt* represent prediction and ground truth, respectively, β is a hyper-parameter and is usually taken to be 1, and |·| represents an absolute value. The parameters $\tau_{x}^{p}$, $\tau_{y}^{p}$ and $\tau_{z}^{p}$ are inferred by the network, and $\tau_{x}^{gt}$, $\tau_{y}^{gt}$, and $\tau_{z}^{gt}$ are obtained from $RT_{gt}$ of Equation (4). 

### 3.4. Postprocessing

#### 3.4.1. Generation of a Calibration Matrix from a Network

Basically, postprocessing is performed to generate the calibration matrix $RT_{pred}$ that is shown in Equation (12). The rotation matrix $R_{pred}$ and translation vector $T_{pred}$ in Equation (12) are generated by the quaternion parameters $q_{0}$, $q_{1}$, $q_{2}$, and $q_{3}$, and translation parameters $\tau_{x}^{p}$, $\tau_{y}^{p}$, and $\tau_{z}^{p}$ inferred from the network we built, as shown in Equations (13) and (14).
(12)$$RT_{pred}=\left[\begin{array}{cc} \begin{array}{c} R_{pred}\\ \begin{array}{ccc} 0 & 0 & 0 \end{array} \end{array} & \begin{array}{c} T_{pred}\\ 1 \end{array} \end{array}\right]$$
(13)$$R_{pred}=\left[\begin{array}{ccc} 1-2\left(q_{2}^{2}+q_{3}^{2}\right) & 2\left(q_{1}q_{2}-q_{0}q_{3}\right) & 2\left(q_{0}q_{2}+q_{1}q_{3}\right)\\ 2\left(q_{1}q_{2}+q_{0}q_{3}\right) & 1-2\left(q_{1}^{2}+q_{3}^{2}\right) & 2\left(q_{2}q_{3}-q_{0}q_{1}\right)\\ 2\left(q_{1}q_{3}-q_{0}q_{2}\right) & 2\left(q_{0}q_{1}+q_{2}q_{3}\right) & 1-2\left(q_{1}^{2}+q_{2}^{2}\right) \end{array}\right]$$
(14)$$T_{pred}={\left[\begin{array}{ccc} \tau_{x}^{p} & \tau_{y}^{p} & \tau_{z}^{p} \end{array}\right]}^{T}$$
(15)$$\theta_{x}^{p}=atan2\left(R_{pred}\left(3,2\right),R_{pred}\left(3,3\right)\right)\;\theta_{y}^{p}=atan2\left(-R_{pred}\left(3,1\right),\;\sqrt[{}]{R_{pred}{\left(3,2\right)}^{2}+R_{pred}{\left(3,3\right)}^{2}}\right)\;\theta_{z}^{p}=atan2\left(R_{pred}\left(2,1\right),R_{pred}\left(1,1\right)\right)$$

Equation (15) shows how to calculate the rotation angle about each of the *x*-, *y*-, and *z*-axes from the rotation matrix $R_{pred}$. In Equation (15), (*r*,*c*) indicates the row index *r* and column index *c* of the matrix $R_{pred}$. The angle calculation described in Equation (15) is used to convert a given rotation matrix into Euler angles.

#### 3.4.2. Calculation of Calibration Error

To evaluate the proposed calibration system, it is necessary to calculate the error of the predicted parameters. For this, we calculate the transformation matrix $RT_{error}$, which contains the errors of the predicted parameters by Equation (16). $RT_{mis}$ and $RT_{online}$ in Equation (16) are calculated by Equations (3) and (17), respectively. In Equation (17), each of $RT_{1}$, $RT_{2}$, $RT_{3}$, $RT_{4}$, and $RT_{5}$ is a calibration matrix predicted by each of the five networks, Net1, Net2, Net3, Net4, and Net5. The calculation of these five matrices is described in detail in 3.4.3. From $RT_{error}$, we calculate the error of the rotation-related parameters using Equation (18) and the error of the translation-related parameters using Equation (19).
(16)$$RT_{error}=RT_{online}\cdot RT_{mis}$$
(17)$$RT_{online}=RT_{5}\cdot RT_{4}\cdot RT_{3}\cdot RT_{2}\cdot RT_{1}$$
(18)$$\theta_{x}^{e}=atan2\left(RT_{error}\left(3,2\right),\;RT_{error}\left(3,3\right)\right)\;\theta_{y}^{e}=atan2\left(-RT_{error}\left(3,1\right),\;\sqrt[{}]{RT_{error}{\left(3,2\right)}^{2}+RT_{error}{\left(3,3\right)}^{2}}\right)\;\theta_{z}^{e}=atan2\left(RT_{error}\left(2,1\right),RT_{error}\left(1,1\right)\right)$$
(19)$$\tau_{x}^{e}=RT_{error}\left(1,4\right)\;\tau_{y}^{e}=RT_{error}\left(2,4\right)\;\tau_{z}^{e}=RT_{error}\left(3,4\right)$$

In Equations (18) and (19), (*r*,*c*) indicates the row index *r* and column index *c* of the matrix $RT_{error}$.

In the KITTI dataset, the rotation angle about the *x*-axis, the rotation angle about the *y*-axis, and the rotation angle about the *z*-axis correspond to pitch, yaw, and roll, respectively. In contrast, in the Oxford dataset, they correspond to roll, pitch, and yaw, respectively. 

#### 3.4.3. Iterative Refinement for Precise Calibration

The training uses all five deviation ranges, but the evaluation of the proposed method is performed with randomly sampled deviations only in Rg1, which is the largest deviation range. Using this sampled deviation, the transformation matrix $RT_{mis}$ is formed as shown in Equations (3), (7), and (8). Then, a point cloud prepared for evaluation is initially transformed using Equation (3). By inputting this transformed point cloud into the trained Net1, the values of parameters that describe translation and rotation are inferred. With these inferred values, we obtain the $RT_{pred}$ of Equation (12). This $RT_{pred}$ becomes $RT_{1}$. We multiply the initial transformed points by this $RT_{1}$ to obtain new transformed points, and we input these new transformed points into the trained Net2 to obtain $RT_{pred}$ from Net2. This new $RT_{pred}$ becomes $RT_{2}$. In this way, the input points to the current network are multiplied by $RT_{pred}$, which is the output of the current network, to obtain new transformed points for use as the input to the next network; this process of obtaining new $RT_{pred}$ by inputting them into the next network is repeated until Net5. For each point cloud prepared for evaluation as described above, a calibration matrix ($RT_{i}$, *i* = 1,···,5) is obtained from each of the five networks, and the final calibration matrix $RT_{online}$ is obtained by multiplying the calibration matrices as shown in Equation (17). The iterative transformation process of the point cloud for evaluation as described above is expressed as follows:(20)$$\hat{P_{1}^{\prime }}=RT_{mis}RT_{init}\hat{P^{T}}$$
(21)$$\hat{P_{i}^{\prime }}=RT_{i-1}\hat{P_{i-1}^{\prime }},\;i=2,\ldots ,5$$

#### 3.4.4. Temporal Filtering for Precise Calibration

Calibration performed with only a single frame can be vulnerable to various forms of noise. According to [25], this problem can be improved by analyzing the results over time. For this purpose, N. Schneider et al. [25] check the distribution of the results over all evaluation frames while maintaining the value of the sampled deviation used for the first frame. They take the median over the whole sequence, which enables the best performance on the test set. They sample the deviations from Rg1. They repeat 100 runs of this experiment, keeping the sampled deviations until all test frames are passed and resampling the deviations at the start of a new run.

It is good to analyze the results obtained over multiple frames. However, applying all the test frames to temporal filtering has a drawback in the context of autonomous driving. In the case of the KITTI dataset, the calibration parameter values are inferred from the results obtained from processing about 4500 frames, which takes a long time. It is also difficult to predict what will happen during this time. Therefore, we reduce the number of frames to use for temporal filtering and randomly determine the start frame for filtering among these frames. We set the bundle size of frames to 100 and performed quantitative analysis by taking the median from 100 results obtained by applying this bundle. The value of parameters from $RT_{online}$ for each frame is obtained using Equations (14) and (15). The basis for setting the bundle size is given in Section 4.3.3.

## 4. Experiments

There are several tasks, such as data preparation in training and evaluation of the proposed calibration system. The KITTI dataset provides images captured with a stereo camera and point clouds acquired using a LiDAR. The dataset consists of 21 sequences (00 to 20) from different scenarios. The Oxford dataset provides point clouds acquired using two LiDARs. In addition, both datasets provide initial calibration parameters and visual odometry information.

We used the KITTI dataset for LiDAR-stereo camera calibration. We referred to the method proposed by Lv et al. [1] in using the 00 sequence (4541 frames) for testing and using the rest (39,011 frames) of the sequences for training. We used the Oxford dataset for LiDAR-LiDAR calibration. Of the many sequences in the Oxford dataset, we used the 2019-01-10-12-32-52 sequence for training and the 2019-01-17-12-32-52 sequence for evaluation. The two LiDARs that were used to build the Oxford dataset were not synchronized. Therefore, we used visual odometry information to synchronize the frames. After the synchronization, the unsynchronized frames were deleted, and our Oxford dataset consisted of 43,130 frames for training and 35,989 frames for evaluation.

We did not apply the same hyper-parameter values to all five networks (Net1 to Net5) because of the large difference in the range of allowable deviations for rotation and translation in Rg1 and Rg5. Because Net5 is trained with Rg5, which has the smallest deviation range, and is applied last in determining the calibration matrix, we trained Net5 using different hyper-parameter values from other networks. Such hyper-parameters included S, *V_r_*, *V_t_*, G, λ_1_, λ_2_, and *B*, which are the length of a side of a voxel, the number of voxels with data among voxels in a voxel space of the reference sensor, the number of voxels with data among voxels in a voxel space of the target sensor, the number of output nodes of the FC_2_ and FC_3_ in the AM, the weight of the loss function $L_{rot}$, the weight of the loss function $L_{trs}$, and the batch size, respectively.

Through the experiments with the Oxford dataset, we observed that data screening is required to enhance the calibration accuracy. The dataset was built with two LiDARs mounted at the left and right corners in front of the roof of a platform vehicle. Figure 2 shows a point cloud for one frame in the Oxford dataset. This point cloud contains points generated by scanning the surface of the platform vehicle by LiDARs. We confirmed that the calibrations performed on point clouds containing these points degrade the calibration accuracy. Therefore, to perform calibration after excluding these points, we set a point removal area to [Horizon: −5–5 m, Vertical: −2–1 m, Depth: −5–5 m] for the target sensor and [Horizon: −1.5–1.5 m, Vertical: −2–1 m, Depth: −2.5–1.5 m] for the reference sensor. Experimental results with respect to this region cropping are provided in Section 4.3.1.

We trained the network for a total of 60 epochs. We initially set the learning rate to 0.0005 and halved it when the epochs reached 30, and we halved it again when the epochs reached 40. The batch size *B* was determined to be within the limits allowed by the memory of the equipment used. We used one NVIDIA GeForce RTX 2080Ti graphic card for all our experiments. Adam [41] was used for model optimization, and hyper-parameters $\beta_{1}$ = 0.9 and $\beta_{2}$ = 0.999 were used.

### 4.1. Evaluation Using the KITTI Dataset

Figure 3 shows a visual representation of the results for performing calibration on the KITTI dataset using the proposed five networks. In this experiment, we transform a point cloud using the calibration matrix inferred from the proposed network and using the ground-truth parameters given in the dataset. We want to show how consistent these two transformation results are. Figure 3a,b show the transformation of a point cloud by randomly sampled deviations from Rg1 and the calibrated parameters given in the KITTI dataset, respectively. The left side of Figure 3c shows the transformation of the point cloud by $RT_{1}$ predicted by the trained Net1. This result looks suitable, but as shown to the right of Figure 3c, it can be seen that the points measured on a thin column were projected to positions that deviated from the column. The effect of iterative refinement appears here. Calibration does not end at Net1 but continues to Net5. Figure 3d shows the transformation of the point cloud by $RT_{online}$ obtained after performing calibration up to Net5. By comparing the result of Figure 3d with the result shown in Figure 3c, we can see that the calibration accuracy is improved: suitable alignment even with the thin column. 

Table 1 presents the average performance of calibrations performed without temporal filtering on 4541 frames for testing on the KITTI dataset. From the results shown in Table 1, we can see the effect of iterative refinement. From Net1 to Net5, the improvements are progressive. Our method achieves an average rotation error of [Roll: 0.024°, Pitch: 0.018°, Yaw: 0.060°] and an average translation error of [X: 0.472 cm, Y: 0.272 cm, Z: 0.448 cm].

Figure 4 shows two examples of error distribution for individual components by means of boxplots. From these experiments, we confirmed that temporal filtering provides suitable calibration results regardless of the amount of arbitrary deviation. The dots shown in Figure 4a,b are both obtained by transforming the same point cloud of the target sensor by randomly sampled deviations from Rg1, but the sampled deviations are different. As can be seen from the boxplots in Figure 4e–h, the distribution of calibration errors was similar despite the large difference in sampled deviations. 

Table 2 shows the calibration results for our method and for the existing CNN-based online calibration methods. From these results, it can be seen that our method achieves the best performance. In addition, when these results are compared with the results shown in Table 1, it can be concluded that our method achieves significant performance improvement through temporal filtering. CalibNet [2] did not specify a frame bundle.

Figure 5 graphically shows the changes in the losses calculated by Equations (10) and (11) for training the proposed networks on the KITTI dataset. In this figure, the green graph shows the results of training with randomly sampled deviations from Rg1, and the pink graph shows the results of training with randomly sampled deviations from Rg5. The horizontal and vertical axes of these graphs represent epochs and loss, respectively. From these graphs, we can observe that the loss reduction decreases from approximately the 30th epoch. This was consistently observed, no matter what deviation range the network was trained on or what hyper-parameters were used. There were similar trends in loss reduction for rotation and translation. Given this situation, we halved the initial learning rate after the 30th epoch of training. Training was performed at a reduced learning rate for 10 epochs after the 30th epoch. After the 40th epoch, we halved the learning rate again. Training continued until the 60th epoch, and the result that produced the smallest training error among the results obtained from the 45th to the 60th epoch was selected as the training result. When Net1 was trained, the hyper-parameters were set as S = 5, (*V_r_*, *V_t_*) = (96, 160), G = 1024, (*λ*_1_, *λ*_2_) = (1, 2), and *B* = 8. When Net5 was trained, the hyper-parameters were set as S = 2.5, (*V_r_*, *V_t_*) = (384, 416), G = 128, (*λ*_1_, *λ*_2_) = (0.5, 5), and *B* = 4. In Figure 5, the training results before the 10th epoch are not shown because the loss was too large.

### 4.2. Evaluation Using the Oxford Dataset

Figure 6 and Figure 7 show the results of performing calibration on the Oxford dataset using the proposed five networks. In these figures, the green dots represent the points obtained by the right LiDAR, which is considered to be the target sensor, and the red dots represent the points obtained by the left LiDAR. Figure 6a,b show the results of the transformation of a point cloud from the target sensor by randomly sampled deviations from Rg1 and calibrated parameters given in the Oxford dataset, respectively. Figure 6c shows the result of the transformation of the point cloud by $RT_{1}$ inferred from the trained Net1. Figure 6d shows the result of the transformation of the point cloud by $RT_{online}$ obtained after performing calibration up to Net5. Similar to the results of the calibration performed using the KITTI dataset, the results of Net1 look suitable, but they are not suitable when compared with the results shown in Figure 6d. The photo on the right side of Figure 6c shows that the green and red dots indicated by an arrow are misaligned. In contrast, the photo on the right side of Figure 6d shows that the green and red dots indicated by an arrow are well aligned. We show through this comparison that calibration accuracy can be improved by the iterative refinement of five networks even without temporal filtering.

Table 3 presents the average performance of calibrations performed without temporal filtering on 35,989 frames for testing in the Oxford dataset. Our method achieves an average rotation error of [Roll: 0.056°, Pitch: 0.029°, Yaw: 0.082°] and an average translation error of [X: 0.520 cm, Y: 0.628 cm, Z: 0.350 cm]. In this experiment, we applied the same hyper-parameters to all five networks. They are S = 5, (*V_r_*, *V_t_*) = (224, 288), G = 1024, (*λ*_1_, *λ*_2_) = (1, 2), and B = 8.

Figure 7 shows two examples of the error distribution of individual components by means of boxplots, as shown in Figure 4. From these experiments, we can see that temporal filtering provides suitable calibration results regardless of the amount of arbitrary deviation, even for LiDAR-LiDAR calibration. The green dots shown in Figure 7a,b are both obtained by transforming the same point cloud of the target sensor with randomly sampled deviations from Rg1, but the sampled deviations are different. As shown in Figure 7e–g, the distribution of calibration errors is similar despite the large difference in sampled deviations. In these experiments, the size of the frame bundle used in the temporal filtering was 100.

Table 4 shows the calibration performance of the proposed method with temporal filtering. Our method achieves a rotation error of less than 0.1° and a translation error of less than 1 cm. By comparing Table 3 and Table 4, it can be seen that temporal filtering achieves a significant improvement in performance.

Figure 8 graphically shows the changes in the losses calculated by Equations (10) and (11) in training the proposed networks with the Oxford dataset. Compared with the results shown in Figure 5, we observed that the results from this experiment were very similar to the experimental results achieved with the KITTI dataset. Therefore, we decided to apply the same training strategy to the KITTI and Oxford datasets. However, the settings of the hyper-parameter values that were applied to the network were different. When Net1 was trained, the hyper-parameters were set as S = 5, (*V_r_*, *V_t_*) = (224, 288), G = 1024, (*λ*_1_, *λ*_2_) = (1, 2), and *B* = 8. When Net5 was trained, the hyper-parameters were set as S = 5, (*V_r_*, *V_t_*) = (224, 288), G = 1024, (*λ*_1_, *λ*_2_) = (0.5, 5), and *B* = 4.

### 4.3. Ablation Studies

#### 4.3.1. Performance According to the Cropped Area of the Oxford Dataset

At the beginning of Section 4, we mentioned the need to eliminate some points in the Oxford dataset that degraded calibration performance. To support this observation, we presented in Table 5 the results of experiments with and without the removal of those points. However, although there is a difference in the calibration performance according to the size of the removed area, it is difficult to theoretically determine the size of the area to be cropped. Table 5 shows the results of the experiments by setting the area to be cut in two ways. Through these experiments, we found that the calibration performed after removing points that caused the performance degradation generally produced better results than the calibration performed without removing those points. These experiments were performed with the trained Net5, and the hyper-parameters were as follows. S = 5, V = (224, 288), G = 1024, λ = (1, 2), and B = 8.

#### 4.3.2. Performance According to the Length of a Voxel Side, S

We conducted experiments to check how the calibration performance changes according to S. Table 6 and Table 7 show the results of these experiments. Table 6 shows the results for the KITTI dataset, and Table 7 shows the results for the Oxford dataset. We performed an evaluation according to S with a combination of Rg1 and Net1 and a combination of Rg5 and Net5. These experiments showed that the calibration performance improved as S became smaller. However, as S became smaller, the computational cost increased, and in some cases, the performance deteriorated. We tried to experiment with fixed values of hyper-parameters other than S, but naturally, as S decreased, the hyper-parameters *V_r_* and *V_t_* increased rapidly. This was a burden on the memory, and thus it was difficult to keep the batch size B at the same value. Therefore, when S was 2.5, B was 4 in the experiment performed on the KITTI dataset, and B was 2 in the experiment performed on the Oxford dataset. However, for S greater than 2.5, B was fixed at 8. In addition, there were cases where the performance deteriorated when S was very small, such as 2.5, which was considered to be the result of a small receptive field in the FEN. Even in the experiments performed on the Oxford dataset, when S was 2.5 in Net1, the training loss diverged near the 5th epoch, so the experiment could no longer be performed. For training on the KITTI dataset, S was set to 2.5 in Net5, and S was set to 5 in Net1 to Net4. However, for training on the Oxford dataset, S was set to 5 for both Net1 and Net5.

#### 4.3.3. Performance According to the Bundle Size of Frames

We conducted experiments to observe how the calibration performance changes according to the bundle size of the frame for temporal filtering. Table 8 and Table 9 show the results of these experiments. Table 8 shows the results for the KITTI dataset, and Table 9 shows the results for the Oxford dataset. We performed the experiments as presented in Section 3.4.3. Because 100 runs had to be performed, the position of the starting frame for each run was predetermined. For each run, we took the median of the values of each of the six parameters associated with rotation and translation inferred from the frames in the bundle, and we calculated the absolute difference between this median and the deviation randomly sampled from Rg1. The error of each parameter shown in Table 8 and Table 9 was obtained by adding up the error of the corresponding parameters calculated for each run and dividing the sum by the number of runs. Through these experiments, we found that temporal filtering using many frames improves the overall calibration performance. However, if we look carefully at the results presented in the two tables, the effect is not shown for all parameters. Considering this observation and the processing time, the bundle size of the frame was set to 100.

## 5. Conclusions

In this paper, we realized a novel approach for online multi-sensor calibration implemented using a voxel-based CNN and 3D convolutional kernels. Our method aims to calibrate between sensors that can measure 3D space. In particular, the voxelization that converts the input 3D point cloud into voxel and the AM introduced to find the correlation of features between the reference and target sensors contributed greatly to the completeness of the proposed method. We demonstrated through experiments that the proposed method can perform both LiDAR-stereo camera calibration and LiDAR-LiDAR calibration. In the calibration of the LiDAR-stereo camera combination, the proposed method showed experimental results that surpassed all existing CNN-based calibration methods for the LiDAR-camera combination. We demonstrated the effects of iterative refinement on the five networks and the effects of temporal filtering through experiments. The proposed method achieved a rotation error of less than 0.1° and a translation error of less than 1 cm on both the KITTI and Oxford datasets.

## Figures and Tables

**Figure 1 sensors-22-06447-f001:**
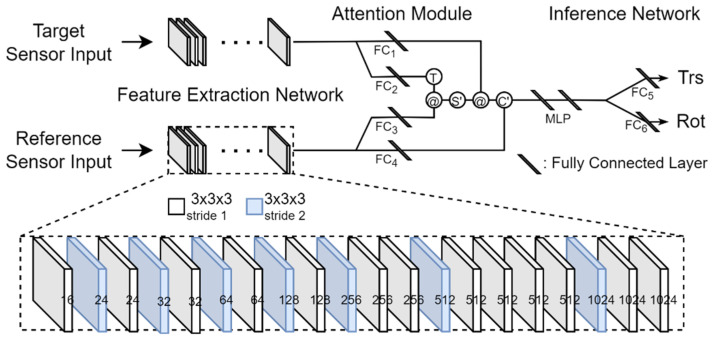
Overall structure of the proposed network. In the attention module, the T within a circle represents the transpose of a matrix; @ within a circle represents a matrix multiplication; S’ within a circle represents the soft max function; C’ within a circle represents concatenation. In the inference network, Trs and Rot represent the translation and rotation parameters predicted by the network, respectively.

**Figure 2 sensors-22-06447-f002:**
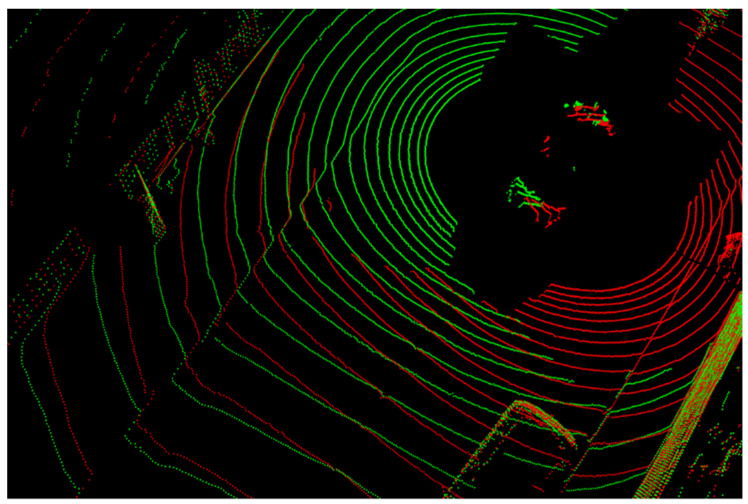
Point cloud constituting one frame in the Oxford dataset. The green dots represent points obtained by the right LiDAR, and the red dots represent the points obtained by the left LiDAR.

**Figure 3 sensors-22-06447-f003:**
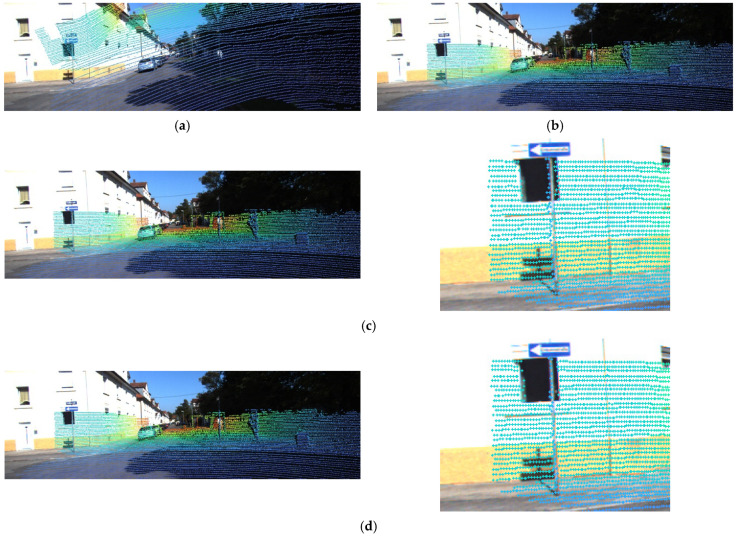
Results of applying the proposed method to a test frame of the KITTI dataset. (**a**) Transformation by randomly sampled deviations. (**b**) Transformation by given calibrated parameters. (**c**) Transformation by $RT_{1}$ inferred from Net1. (**d**) Transformation by $RT_{online}$ obtained from iterative refinement by five networks.

**Figure 4 sensors-22-06447-f004:**
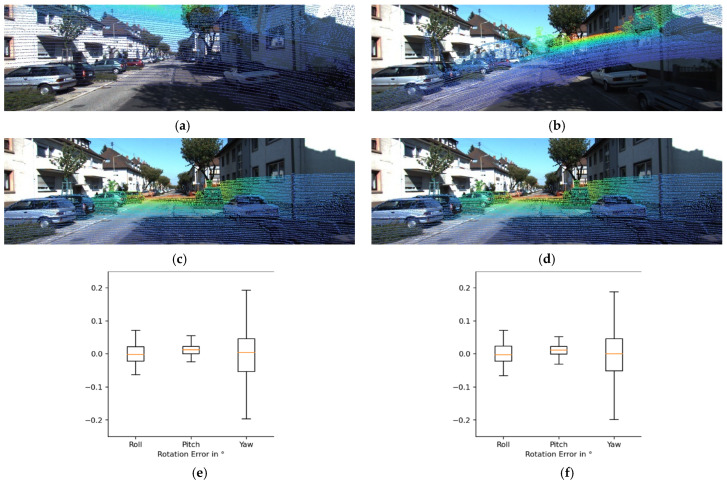
Calibration results and error distribution when temporal filtering was applied. (**a**) Transformation by randomly sampled deviation from Rg1. (**b**) Transformation by randomly sampled deviation from Rg1. (**c**) Calibration results from random deviations shown in (**a**). (**d**) Calibration results from random deviations shown in (**b**). (**e**) Rotation error for the results shown in (**c**). (**f**) Rotation error for the results shown in (**d**). (**g**) Translation error for the results shown in (**c**). (**h**) Translation error for the results shown in (**d**).

**Figure 5 sensors-22-06447-f005:**
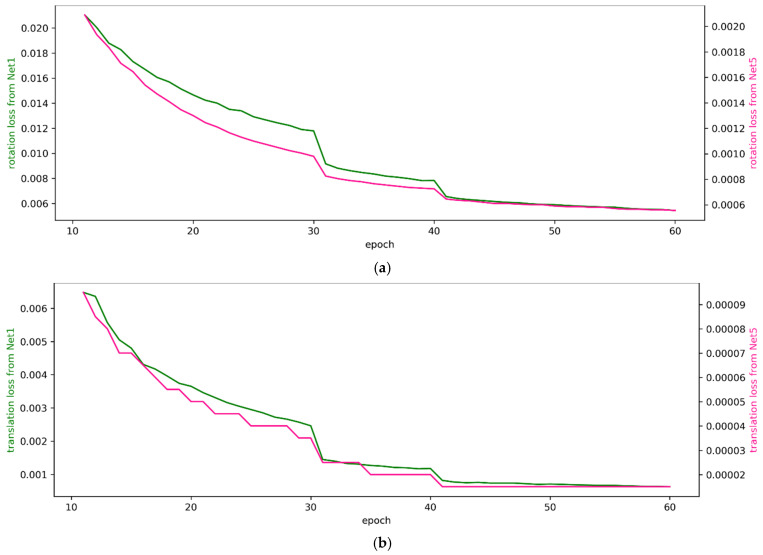
Changes in loss calculated during the training of Net1 and Net5 on the KITTI dataset. (**a**) $L_{rot}$ calculated using Equation (10). (**b**) $L_{trs}$ calculated using Equation (11).

**Figure 6 sensors-22-06447-f006:**
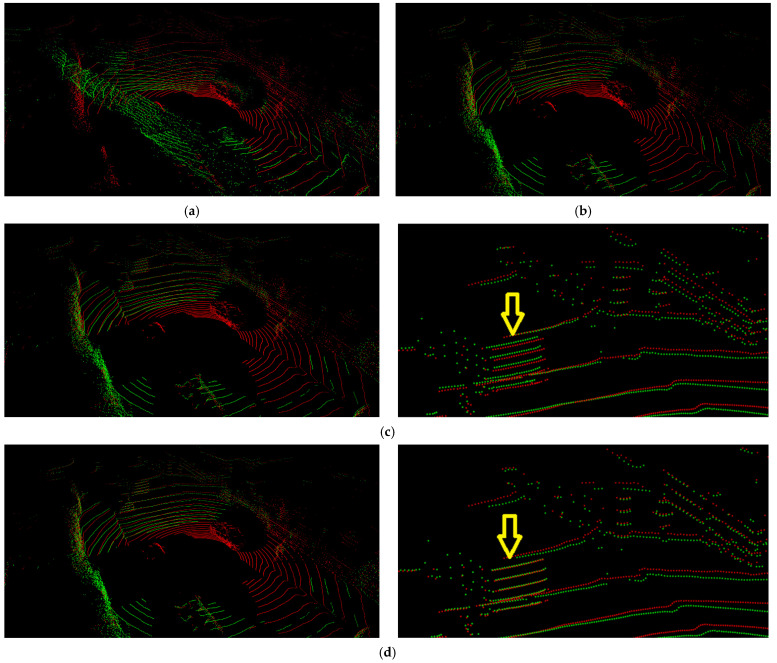
Results of applying the proposed method to a test frame of the Oxford dataset. (**a**) Transformation by randomly sampled deviations. (**b**) Transformation by given calibrated parameters. (**c**) Transformation by $RT_{1}$ inferred from Net1. (**d**) Transformation by $RT_{online}$ obtained from iterative refinement by five networks.

**Figure 7 sensors-22-06447-f007:**
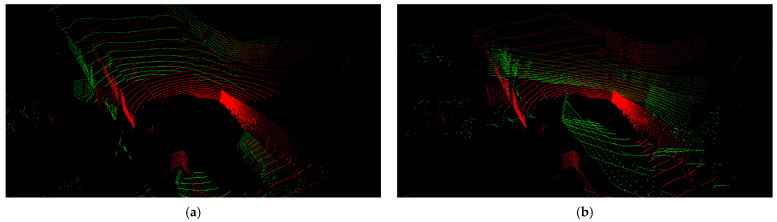
Calibration results and error distribution when temporal filtering was applied to the Oxford dataset. (**a**) Transformation by randomly sampled deviations from Rg1. (**b**) Transformation by randomly sampled deviations from Rg1. (**c**) Calibration results from random deviations shown in (**a**). (**d**) Calibration results from random deviations shown in (**b**). (**e**) Rotation error for the results shown in (**c**). (**f**) Rotation error for the results shown in (**d**). (**g**) Translation error for the results shown in (**c**). (**h**) Translation error for the results shown in (**d**).

**Figure 8 sensors-22-06447-f008:**
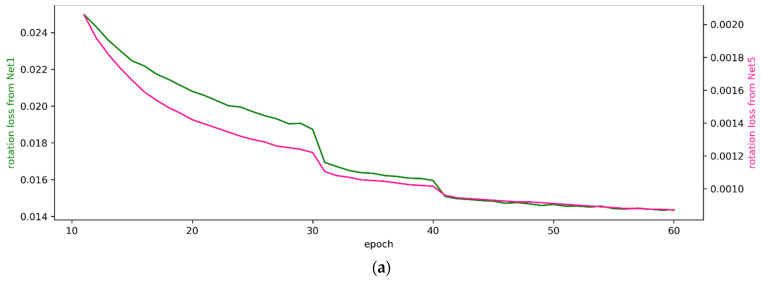
Changes in calculated losses during Net1 and Net5 training on the Oxford dataset. (**a**) Rotation loss $L_{rot}$ calculated using Equation (10). (**b**) Translation loss $L_{trs}$ calculated using Equation (11).

**Table 1 sensors-22-06447-t001:** Quantitative results of calibration performed on the KITTI dataset without temporal filtering. See footnotes ^1,2^ for hyper-parameter settings.

Refinement Stage	Rotation Error (°)			Translation Error (cm)		
	Roll	Pitch	Yaw	X	Y	Z
After Net1 ^1^	0.182	0.110	0.386	2.393	1.205	1.781
After Net2 ^1^	0.112	0.068	0.176	1.513	1.356	1.663
After Net3 ^1^	0.071	0.046	0.134	1.119	0.709	1.027
After Net4 ^1^	0.039	0.024	0.088	0.750	0.428	0.735
After Net5 ^2^	0.024	0.018	0.060	0.472	0.272	0.448

^1^ S = 5, (*V_r_*, *V_t_*) = (96, 160), G = 1024, (*λ*_1_, *λ*_2_) = (1, 2), *B* = 8. ^2^ S = 2.5, (*V_r_*, *V_t_*) = (384, 416), G = 128, (*λ*_1_, *λ*_2_) = (0.5, 5), *B* = 4.

**Table 2 sensors-22-06447-t002:** Comparison of calibration performance between our method and other CNN-based methods.

Method	Range	Bundle Size of Frame	Rotation Error (°)			Translation Error (cm)		
			Roll	Pitch	Yaw	X	Y	Z
RegNet [25]	Rg1	4541	0.24	0.25	0.36	7	7	4
CalibNet [2]	(±10°, ±0.2 m)	-	0.18	0.9	0.15	4.2	1.6	7.22
LCCNet [1]	Rg1	4541	0.020	0.012	0.019	0.262	0.271	0.357
Ours	Rg1	4541	0.002	0.011	0.004	0.183	0.068	0.183

**Table 3 sensors-22-06447-t003:** Quantitative results of calibration performed on the Oxford dataset without temporal filtering.

Refinement Stage	Rotation Error (°)			Translation Error (cm)		
	Roll	Pitch	Yaw	X	Y	Z
After Net1	0.302	0.223	0.370	3.052	4.440	3.603
After Net2	0.249	0.262	0.266	1.048	2.155	2.240
After Net3	0.136	0.068	0.099	1.469	1.191	1.348
After Net4	0.072	0.036	0.073	0.632	0.809	0.985
After Net5	0.056	0.029	0.082	0.520	0.628	0.350

**Table 4 sensors-22-06447-t004:** Quantitative results of calibration on Oxford dataset with temporal filtering.

Method	Range	Bundle Size of Frame	Rotation Error (°)			Translation Error (cm)		
			Roll	Pitch	Yaw	X	Y	Z
Ours	Rg1	100	0.035	0.017	0.060	0.277	0.305	0.247

**Table 5 sensors-22-06447-t005:** Comparison of calibration performance according to the cropped area on the Oxford dataset.

Size of Area to be Cropped [Horizon, Vertical, Depth]	Rotation Error (°)			Translation Error (cm)		
	Roll	Pitch	Yaw	X	Y	Z
N/A	0.038	0.030	0.070	1.922	0.868	0.476
[−5~5 m, −2–1 m, −5~5 m]	0.033	0.027	0.062	0.538	0.668	0.564
[−10~10 m, −2–1 m, −10~10 m]	0.033	0.025	0.054	0.490	1.109	0.496

**Table 6 sensors-22-06447-t006:** Comparison of calibration performance according to S on the KITTI dataset.

Hyper-Parameter Setting	Rotation Error (°)			Translation Error (cm)		
	Roll	Pitch	Yaw	X	Y	Z
For the Combination of Net1 and Rg1						
S=10, V=(32,48), G=1024, λ = (1, 2), B = 8	0.228	0.166	0.421	3.103	1.681	2.155
S=7.5, V=(48,72), G=1024, λ = (1, 2), B = 8	0.199	0.199	0.429	2.881	1.613	2.514
S=5, V=(96,160), G=1024, λ = (1, 2), B = 8	0.182	0.110	0.386	2.393	1.205	1.781
S=2.5, V=(384,416), G=128, λ = (1, 2), B = 4	0.295	0.206	0.595	4.489	2.288	2.840
For the Combination of Net5 and Rg5						
S=10, V=(32,48), G=1024, λ = (1, 2), B = 8	0.344	0.019	0.063	0.778	0.429	0.887
S=7.5, V=(48,72), G=1024, λ = (1, 2), B = 8	0.030	0.020	0.059	0.646	0.487	0.776
S=5, V=(96,160), G=1024, λ = (1, 2), B = 8	0.028	0.017	0.070	0.610	0.363	0.702
S=2.5, V=(384,416), G=128, λ = (1, 2), B = 4	0.023	0.016	0.045	0.450	0.312	0.537

**Table 7 sensors-22-06447-t007:** Comparison of calibration performance according to S on the Oxford dataset.

Hyper-Parameter Setting	Rotation Error (°)			Translation Error (cm)		
	Roll	Pitch	Yaw	X	Y	Z
For the Combination of Net1 and Rg1						
S=10, V=(76,96), G=1024, λ = (1, 2), B = 8	0.382	0.328	0.436	2.606	8.114	2.881
S=7.5, V=(96,160), G=1024, λ = (1, 2), B = 8	0.415	0.263	0.433	3.542	7.574	4.151
S=5, V=(224,288), G=1024, λ = (1, 2), B = 8	0.302	0.223	0.370	3.052	4.440	3.603
S=2.5, V=(608,608), G=128, λ = (1, 2), B = 2	-	-	-	-	-	-
For the Combination of Net5 and Rg5						
S=10, V=(76,96), G=1024, λ = (1, 2), B = 8	0.046	0.031	0.085	0.626	1.431	0.610
S=7.5, V=(96,160), G=1024, λ = (1, 2), B = 8	0.031	0.228	0.535	0.448	1.357	0.457
S=5, V=(224,288), G=1024, λ = (1, 2), B = 8	0.033	0.027	0.062	0.538	0.668	0.564
S=2.5, V=(608,608), G=128, λ = (1, 2), B = 2	0.036	0.025	0.057	0.552	0.699	0.539

**Table 8 sensors-22-06447-t008:** Comparison of calibration performance according to the bundle size of frames for temporal filtering on the KITTI dataset.

Bundle Size of Frames	Rotation Error (°)			Translation Error (cm)		
	Roll	Pitch	Yaw	X	Y	Z
1	0.024	0.017	0.057	0.414	0.257	0.395
10	0.009	0.013	0.018	0.210	0.102	0.245
25	0.006	0.011	0.013	0.176	0.080	0.197
50	0.004	0.011	0.008	0.170	0.070	0.190
100	0.003	0.011	0.006	0.175	0.069	0.195

**Table 9 sensors-22-06447-t009:** Comparison of calibration performance according to the bundle size of frame for temporal filtering on the Oxford dataset.

Bundle Size of Frames	Rotation Error (°)			Translation Error (cm)		
	Roll	Pitch	Yaw	X	Y	Z
1	0.055	0.028	0.080	0.536	0.532	0.330
10	0.049	0.024	0.066	0.363	0.335	0.305
25	0.044	0.022	0.066	0.334	0.303	0.272
50	0.039	0.019	0.065	0.290	0.286	0.269
100	0.035	0.017	0.060	0.277	0.305	0.247

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
