# Peer review of "Online Self-Calibration of 3D Measurement Sensors Using a Voxel-Based Network"

_sensors, 2022, doi:10.3390/s22176447_

Round 1
Reviewer 1 Report
I have reviewed the paper and here are my comments
1. The paper might be translated and copied from the author's own thesis.. that is why the author used .....This Chapter described..... Line 161.
2. Few of the self-calibrated schemes are presented in the related work section. But I wondered why the author don't described and cite the well-known calibration schemes like 10.1109/ICCSN.2016.7587200. I suggest improving related work section and citing these work
3. How the stereo matching has been formed.
4. Eq 6: is the relative displacement in the form of a scaler or vector. Or is it a coordinates values? The representation doesnt clearly described.
5. Understood that the sensors provide 3D points, but how: is the beacon involved in it? Is the sensor broadcast beacons for obtaining those points in voxelization process?
6. What is the motivation behind the proposed scheme? what is the novelty? Line 293... how robust calibration.
7. Line 302: should place a meaningful heading.
8. Results section is reasonably good.
Reviewer 2 Report
In this study, the authors proposed a CNN-based multi-sensor online self-calibration algorithm, which used a LiDAR and stereo camera, one of them as a reference. It used five networks that process voxel information, and improved calibration accuracy through iterative refinement. The proposed method was evaluated with KITTI and Oxford datasets, showed a rotation error of less than 0.1° and a translation error of less than 1 cm.
The manuscript was well written, where the formula and procedures of the technique were carefully described in detail, therefore it showed some value for developing novel autonomous driving and other applications.
However, the timescales of the method in different evaluation experiments were not give, as well as a comparison to other existing methods for the same tasks. And they have not tested the method with real data taken from a dynamic street, road or moving subjects in their city. By adding in these data, this manuscript is acceptable for publication.
Round 2
Reviewer 2 Report
Understand the limitation.